# Switch-NeRF: Learning Scene Decomposition with Mixture of Experts for Large-scale Neural Radiance Fields

**Zhenxing Mi & Dan Xu**
Department of Computer Science and Engineering
The Hong Kong University of Science and Technology (HKUST)
Clear Water Bay, Kowloon, Hong Kong
`zmiaa@connect.ust.hk, danxu@cse.ust.hk`

## Abstract

The Neural Radiance Fields (NeRF) have been recently applied to reconstruct building-scale and even city-scale scenes. To model a large-scale scene efficiently, a dominant strategy is to employ a divide-and-conquer paradigm via performing scene decomposition, which decomposes a complex scene into parts that are further processed by different sub-networks. Existing large-scale NeRFs mainly use heuristic hand-crafted scene decomposition, with regular 3D-distance-based or physical-street-block-based schemes. Although achieving promising results, the hand-crafted schemes limit the capabilities of NeRF in large-scale scene modeling in several aspects. Manually designing a universal scene decomposition rule for different complex scenes is challenging, leading to adaptation issues for different scenarios. The decomposition procedure is not learnable, hindering the network from jointly optimizing the scene decomposition and the radiance fields in an end-to-end manner. The different sub-networks are typically optimized independently, and thus hand-crafted rules are required to composite them to achieve a better consistency. To tackle these issues, we propose Switch-NeRF, a novel end-to-end large-scale NeRF with learning-based scene decomposition. We design a gating network to dispatch 3D points to different NeRF sub-networks. The gating network can be optimized together with the NeRF sub-networks for different scene partitions, by a design with the Sparsely Gated Mixture of Experts (MoE). The outputs from different sub-networks can also be fused in a learnable way in the unified framework to effectively guarantee the consistency of the whole scene. Furthermore, the proposed MoE-based Switch-NeRF model is carefully implemented and optimized to achieve both high-fidelity scene reconstruction and efficient computation. Our method establishes clear state-of-the-art performances on several large-scale datasets. To the best of our knowledge, we are the first to propose an applicable end-to-end sparse NeRF network with learning-based decomposition for large-scale scenes. Codes are released at `https://github.com/MiZhenxing/Switch-NeRF`.

## 1 Introduction

The Neural Radiance Fields (NeRF) method (Mildenhall et al., 2020) has gathered wide popularity in novel-view synthesis and 3D reconstruction due to its high quality and simplicity. It encodes a 3D scene from multiple 2D posed images. The original NeRF typically targets small scenes or objects, while in real-world applications such as autonomous driving and augmented reality (AR) / virtual reality (VR), building NeRF models to effectively handle large-scale scenes is critically important.

The problem of a large-scale NeRF is that more data typically requires a higher network capacity (number of network parameters). A naïve solution is to densely increase the network width and depth. However, this will also greatly increase the computation for each sample and is harder to optimize. A more applicable network should have a large capacity while maintaining almost constant computational cost for each sample. Therefore, building an applicable large-scale NeRF can be

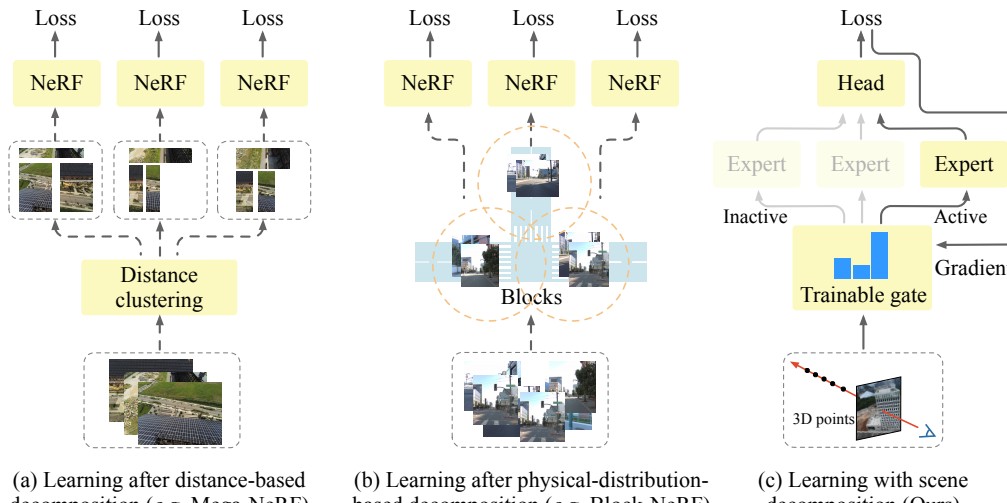

(a) Learning after distance-based decomposition (*e.g.* Mega-NeRF)

(b) Learning after physical-distribution-based decomposition (*e.g.* Block-NeRF)

(c) Learning with scene decomposition (Ours)

Figure 1: Different kinds of decomposition methods. The dot lines mean non-differentiable operations. The solid lines mean differentiable operations that can be trained by back-propagation. The Mega-NeRF (Turki et al., 2022) clusters pixels by 3D sampling distances to centroids in training. The Block-NeRF (Tancik et al., 2022) clusters images by dividing the whole scene according to street blocks. The sub-networks in both methods are trained separately. Our Switch-NeRF learns to decompose the 3D points by a trainable gating network and the whole network is trained end-to-end.

considered as building a sparse neural network. The core of the design is to select different network parameters (*i.e.* sub-networks) for different inputs. This procedure can be formulated as a scene decomposition problem in the NeRF task. Each sub-network handles a different part of the scene.

Along the scene decomposition and learning a sparse neural network, recent Mega-NeRF (Turki et al., 2022) and Block-NeRF (Tancik et al., 2022) have extended NeRF to building-scale and even city-scale scenes based on heuristic hand-crafted scene decomposition. As depicted in Fig. 1, the Mega-NeRF and Block-NeRF simply use 3D sampling distances or street blocks to decompose the scene and train different NeRF models separately. With promising results on large-scale scenes, their hand-crafted scene decomposition methods still lead to several issues. The large-scale scenes are essentially complex and irregular. Designing a universal scene decomposition rule for different scenes is extremely challenging in a hand-crafted way. This accordingly brings adaptation issues for distinct scenarios in the real world. Hand-crafted rules require rich priors of the target scene, such as the structure of the scene, to deploy the partition centroids as in Mega-NeRF and the physical distribution of the scene images as in Block-NeRF. These priors may not be available in practical applications. The hand-crafted decomposition is not learnable, hindering the network from jointly optimizing the scene decomposition and the radiance fields in an end-to-end manner. The gaps between the decomposition, composition and NeRF optimization may lead to sub-optimal results. Besides, the different sub-networks are typically trained separately, leading to possible inconsistency among different sub-networks. To handle this problem, they usually set overlapping among adjacent partitions in training and use hand-crafted rules in inference to composite results from different sub-networks. (Tancik et al., 2022; Turki et al., 2022).

To address above-mentioned issues, in this paper, we make the following contributions.

**An end-to-end framework for joint learning of scene decomposition and NeRF.** We present Switch-NeRF, an end-to-end sparse neural network framework, which jointly learns the scene decomposition and NeRF. As shown in Fig. 1c, we propose a learnable gating network for scene decomposition. It dynamically selects and sparsely activates a sub-network for each 3D point. The overall network is trained end-to-end without any heuristic intervention. We do not require any priors of the 3D scene shape or the distribution of scene images, leading to a generic framework for large-scale scenes. Since the selection of sub-networks in training is a discrete operation, a critical problem is how to back-propagate gradients into the gating network. We use the strategy from the Sparsely-Gated Mixture-of-Experts (MoE) (Shazeer et al., 2017) to deal with this problem. We structure our sub-networks as NeRF experts for different scene partitions. 3D points are dispatched into different NeRF experts based on the gating network. Besides the gating network, we also de-

sign a head to unify the predictions of multiple NeRF experts, which aligns the high-level implicit features from different NeRF experts to effectively address the inconsistency problem.

**Efficient network design and implementation.** With the framework design of Switch-NeRF, however, optimizing and implementing it efficiently and stably is not trivial. In NeRF, the number of 3D points in a forward pass is orders of magnitude larger than that of the input tokens in other NLP and vision tasks within an MoE framework. Dispatching samples to different NeRF experts inevitably introduces large computation and memory usage. Therefore, we consider dispatching 3D points only once with an effective gating network, and we design a deeper gating network to guarantee enough parameters to boost the accuracy of the scene rendering. Another common design in MoE implementations (Hwang et al., 2022) is to define a capacity factor to limit the number of tokens dispatched to each expert. This dynamically drops overflow 3D points for our NeRF experts. It works well when training the network but brings a large influence on the testing accuracy. To address this issue, we implement a full dispatch operation by CUDA based on Tutel (Hwang et al., 2022) to significantly improve the testing performance, while avoiding unnecessary memory allocation.

**High-quality results.** Extensive experiments are conducted on challenging benchmarks with large-scale scenes. Qualitative results demonstrate that our network can learn reasonable decomposition of large-scale complex scenes. Our model also establishes state-of-the-art performances. It shows clearly more superior results with much less network parameters compared to those hand-crafted decomposition counterparts.

## 2 RELATED WORK

**Neural Radiance Field**. The Neural Radiance Field (NeRF) is proposed by Mildenhall et al. (2020) to use volumetric rendering for novel view synthesis from posed images. It encodes a 3D scene into a multilayer perceptron (MLP), which is simple and requires very limited priors of the scene. Due to the success of NeRF in high-quality rendering and 3D reasoning, many works have been proposed to improve its efficiency (Reiser et al., 2021; Yu et al., 2021; Müller et al., 2022), accuracy (Barron et al., 2021; Verbin et al., 2022) and apply it to challenging scenes (Zhang et al., 2020; Martin-Brualla et al., 2021; Xiangli et al., 2022) and 3D reconstruction tasks (Wang et al., 2021).

We pay more attention to the closely related works, *i.e.* NeRF methods for large-scale scenes. As shown in Fig. 1a, Mega-NeRF (Turki et al., 2022) proposes to use a simple 3D distance-based method to cluster training pixels into parts that can be trained separately by different NeRF models. It samples centroids uniformly in the 3D scene and groups 3D points in testing. Block-NeRF (Tancik et al., 2022) proposes to scale NeRF to city-level scenes. As shown in Fig. 1b, it divides the whole scene based on physical distribution of the scene images, *i.e.* partitioning through street blocks. The scene images in different street blocks are trained separately by sub-networks. In the testing, both methods consider offline fusion of prediction results from different sub-networks. In contrast to Mega-NeRF and Block-NeRF which use hand-craft scene decomposition, our Switch-NeRF jointly learns the scene decomposition and a large-scale NeRF in an end-to-end manner. Our method is not dependent on any priors of the 3D shape and the physical image distribution of a target scene. Therefore, our method is more generic for arbitrary large-scale scenes.

**Mixture of Experts (MoE)**. Modern MoE methods mainly follow the work of Shazeer et al. (2017). It proposes a Sparsely-Gated-Mixture-of-Experts layer in place of the feed-forward network (*i.e.* FFN or MLP) in a language model. It designs a vanilla Top-$k$ gating network to dispatch samples into $k$ experts, and proposes an auxiliary loss for balancing the training of different experts. The MoE has been widely used in Natural Language Processing (NLP) (Lepikhin et al., 2021; Fedus et al., 2022) and Vision (Riquelme et al., 2021). Switch Transformer (Fedus et al., 2022) suggests that Top-2 gating is not necessary. It trains a network of high quality with Top-1 gating to largely reduce the dispatch computation and communication. Besides, different gating mechanisms are also proposed, such as Hash Routing (Roller et al., 2021) and BASE (Lewis et al., 2021). There have been popular implementations of MoE in Mesh-TensorFlow (Shazeer et al., 2018), Deepspeed (Rajbhandari et al., 2022) and Tutel (Hwang et al., 2022), which focus on improving large-scale distributed training. To the best of our knowledge, none of the existing works considers developing a Mixture-of-NeRF-Experts (MoNE) for large-scale scenes. We design an effective structure of MoNE for jointly learning scene decomposition and NeRF, and also improve the efficiency of MoNE by handling the issue of dispatching large-scale 3D points to different NeRF experts.

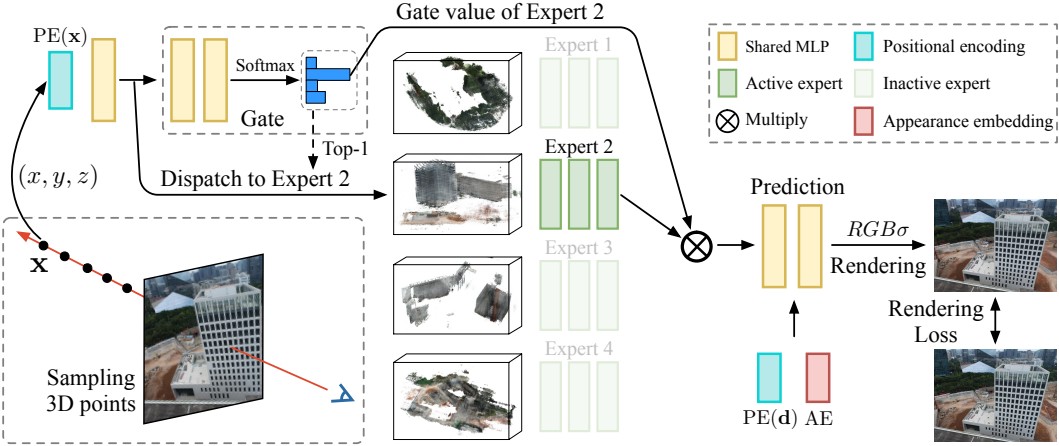

Figure 2: The framework our Switch-NeRF. A 3D point **x** will first go through a gating network and then be dispatched to only one expert according to the gating network output. The expert output is multiplied by the corresponding gate value and sent to a head for density $\sigma$ and color **c** prediction with direction **d** and appearance embedding. The rendering loss is used for supervision. The images on the left of each expert are the visualization of 3D radiance fields handled by different experts.

## 3 SWITCH-NERF

Our Switch-NeRF is a Sparse Neural Radiance Field network targeting large-scale scenes. A framework overview of Switch-NeRF is depicted in Fig. 2. We represent a large-scale 3D scene as a sparse 5D radiance function $F_\Theta : (\mathbf{x}, \mathbf{d}) \rightarrow (\mathbf{c}, \sigma)$, where $(\mathbf{x}, \mathbf{d})$ are a 3D point and its view direction, $(\mathbf{c}, \sigma)$ are predicted color and density, $\Theta$ represents network parameters. $F_\Theta$ sparsely activates only a part of its parameters for an input **x** each time. The overall structure of our network mainly consists of a gating network $G$, a set of $n$ experts $\{E_i\}_{i=1}^n$, and a shared prediction head $H$ for generating $\sigma$ and **c**. The actual inputs of our network are the positional encoding $(PE(\mathbf{x}), PE(\mathbf{d}))$ for $(\mathbf{x}, \mathbf{d})$ as in the vanilla NeRF (Mildenhall et al., 2020). We omit $PE(\cdot)$ in our following equations for simplicity.

Given an input $(\mathbf{x}, \mathbf{d})$, we first send **x** into the gating network and obtain the gate values $G(\mathbf{x})$. Then, we apply a Top-1 operation on $G(\mathbf{x})$ to determine which expert should be activated. As shown in Fig. 2, only one expert (*i.e.* Expert 2) is activated. The point **x** will be dispatched to this selected expert. Other experts do not participate in the processing of **x**. The output feature of the expert $E(\mathbf{x})$ will be multiplied by the gate value corresponding to this expert. This makes the gating network be trained jointly with the expert networks. After that, the feature is used to predict the density $\sigma$ and color **c** together with **d** and the appearance embedding AE. In the next, we first introduce details of the trainable gating and our gating network architecture in Sec. 3.1. Then we introduce our expert and head network architectures in Sec. 3.2. We discuss the capacity factor and full dispatch in Sec. 3.3. We finally formulate the rendering procedure and our loss functions in Sec. 3.4.

### 3.1 SPARSE GATING IN SWITCH-NERF

The sparse gating network plays an important role in our Switch-NeRF because it determines the optimization routes of different NeRF experts. In Switch-NeRF, we only consider one gating network because there is a very large number of 3D points that require gating and dispatching in NeRF optimization. Multiple gating operations will largely decrease the training and testing speed.

**Trainable gating in Switch-NeRF.** The trainable gating in our network follows the mechanism in the MoE method Switch Transformer (Shazeer et al., 2017). The gate values $G(\mathbf{x})$ is a vector of $n$-dimensions normalized via Softmax, in which $G(\mathbf{x})_i$ represents the probability of selecting the $i$-th NeRF expert. We apply a Top-1 function on $G(\mathbf{x})$ to sparsely select only 1 expert $E_s$ from a set of NeRF experts, *i.e.* $\{E_i\}_{i=1}^n$, for each 3D point. The input **x** will be dispatched into the selected expert $E_s$ and obtain an output $E_s(\mathbf{x})$. The final output $\tilde{E}(\mathbf{x})$ the output of $E_s$ multiplied by the corresponding gate value:

$$\tilde{E}(\mathbf{x}) = G(\mathbf{x})_s E_s(\mathbf{x}). \tag{1}$$

As the predicted gate values are multiplied to the corresponding outputs of NeRF experts, the gating network can be optimized together with the NeRF experts in the backward pass. This

makes the network able to directly learn scene decomposition during network training. Our network structure is highly sparse because we use Top-1 to select only one experts for each sample.

**Gating network architecture.** In previous MoE methods (Shazeer et al., 2017; Lepikhin et al., 2021), the gating network is typically a simple linear mapping. This choice is reasonable as they usually have multiple MoE layers. The input of gating networks in deep MoE layers can be high-level features from previous layers. Therefore, their gating networks share more information from the main network. Let the original input be the 3D point **x**. Typically **x** will first go through a sub-network $S$ with several layers and learn an internal feature $S(\mathbf{x})$. The real gating operation of the deep MoE layers can be written as:

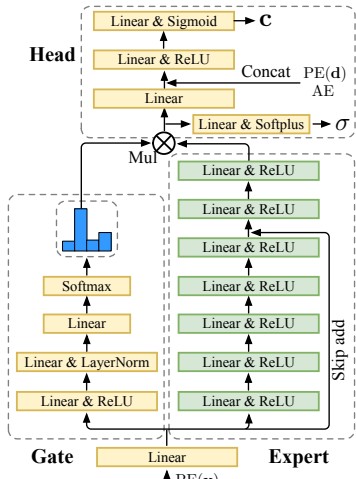

$$G(\mathbf{x}) = \text{Softmax}(\text{Linear}(S(\mathbf{x}))). \tag{2}$$

Therefore, the gating networks in these methods actually share an $S$ network from the main network.

Considering that we only use one gate network in our Switch-NeRF for efficiency, allocating more parameters for our gating network is necessary to learn powerful gating. We can put the gating network closer to the prediction head. It will share more parameters with the main network. However, this will also make each sample point share more layers before the gating network. In this case, the capacity of the whole network shrinks, as the network capacity is controlled by the number of layers of each unshared expert. Therefore, as shown in the Fig. 3, we put the gating network at the beginning of Switch-NeRF to maximize

Figure 3: The architectures of our gating, one expert and the head networks. It shows the forward pass of a point. It goes through the gating network and a selected expert, and is passed to a shared prediction head.

the layer numbers of the expert networks. In order to allocate more parameters to the gating network, we use a shallow MLP instead of a linear mapping. Our gating network $G(\mathbf{x})$ consists of 4 Linear layers and 1 LayerNorm as shown in Fig. 3. Our design balances the efficiency, network sparsity and the number of parameters in the gating network.

## 3.2 Expert and Head Networks

**NeRF Expert Network.** The set of $n$ NeRF experts $\{E_i\}_{i=1}^n$ in Switch-NeRF contains most of the network parameters. As shown in Fig. 3, each expert in Switch-NeRF is a deep MLP with a skip connection. The structure and depth of the experts are aligned with the main structure of vanilla NeRF. Each expert only processes a part of the 3D points, determined by the gating network. The output of the selected expert is multiplied by the corresponding gate value to obtain a feature vector $\tilde{E}(\mathbf{x})$. This feature vector is then used to predict density $\sigma$ and the direction-dependent color **c**. By increasing the number of NeRF experts $n$, we can easily scale the network's capacity.

**Unified NeRF Head.** The head network $H$ is designed for the final predictions of each input sample. It is shared for all the samples, as shown in Fig. 3. After obtaining the expert output $\tilde{E}(\mathbf{x})$, we use a Linear layer with a Softplus activation (Zheng et al., 2015) to predict $\sigma$. The use of Softplus follows Mip-NeRF (Barron et al., 2021) for stable prediction of $\sigma$. Then, $\tilde{E}(\mathbf{x})$ goes through a Linear layer and is concatenated with PE(**d**) and an appearance embedding AE. The color **c** is predicted from the concatenated feature by an MLP. The appearance embedding AE is a trainable vector to capture image-level photometric and environmental variations (Martin-Brualla et al., 2021).

## 3.3 Capacity Factor and Full Dispatch

**Capacity factor for training.** In our Switch-NeRF, to efficiently dispatch 3D points to different experts is important. Einops-based dispatch (Lepikhin et al., 2021) causes memory overflow due to the large number of 3D points. In our training, we use the CUDA-based fast dispatch in Tutel (Hwang et al., 2022). Following previous MoE methods (Lepikhin et al., 2021), we set a capacity factor for each NeRF expert in the training. It caps the number of sample points dispatched to each NeRF expert, leading to uniform tensor shapes, balanced computation and communication. Let $B$ be the batch size of the whole network, $C_f$ be a *capacity factor* and $n$ be the number of NeRF experts. Then the maximum number of sample points going into each NeRF expert is $B_e = \text{ceil}(\frac{kBC_f}{n})$.

The dispatch with a capacity factor can be called a uniform dispatch. Fig. 4 shows the uniform dispatch with $C_f = 1.0$. Overflow points are dropped. If the capacity is not fully used, it will be zero-padded. A larger capacity factor decreases the dropping ratio but increases the memory and computation. In our network, we set the capacity factor to 1.0 without requiring extra memory. We use the Batch Prioritized Routing (Riquelme et al., 2021) to improve the training with lower expert capacity.

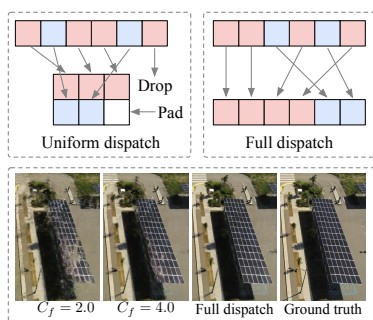

**Full dispatch for testing.** Previous MoE methods usually use the uniform dispatch for both the training and testing (Lepikhin et al., 2021; Fedus et al., 2022), which inevitably drops sample points. In our Switch-NeRF, although the uniform dispatch works well in the training, we observe that dropping sample points can significantly decrease the test accuracy. A possible reason is that we do not use stacked MoE layers and skip connections between them as previous MoE methods usually do, to maintain the network efficiency. To improve the testing accuracy, we implement an efficient full dispatch strategy based on Pytorch, CUDA, and Tutel. It can dispatch all the points

Figure 4: In the uniform dispatch, each expert has the same tensor shapes. It will drop overflow tokens and perform padding. The full dispatch makes sure each point will be processed by an expert. The images rendered by uniform dispatch have apparent artifacts.

to their corresponding expert with only slight memory increase. A description of the strategy is depicted in Fig. 4. With this efficient full dispatch, we can largely improve the test accuracy.

### 3.4 Volume Rendering and Losses

Our volume rendering procedure follows the vanilla NeRF (Mildenhall et al., 2020). The training data of Switch-NeRF consists of multiple posed images. For each pixel in the training images, we back-project a camera ray $\mathbf{r}$ into a 3D space. A set of $N$ samples are sampled along the ray. For each sample $(\mathbf{x}, \mathbf{d})$, our network $F_\Theta$ predicts a volume density $\sigma$ and a color $\mathbf{c} = (r, g, b)$. Let $\delta_i$ be the distance between adjacent points. The expected color $\hat{C}(\mathbf{r})$ of this pixel is synthesized along the ray by the volume rendering function in NeRF (Mildenhall et al., 2020):

$$\hat{C}(\mathbf{r}) = \sum_{i=1}^{N} T_i (1 - \exp(-\sigma_i \delta_i)) \mathbf{c}_i, \text{ where } T_i = \exp(-\sum_{j=1}^{i-1} \sigma_j \delta_j), \tag{3}$$

**Rendering loss.** The main loss of Switch-NeRF is the rendering loss $L_r$. After rendering the color $\hat{C}(\mathbf{r})$ of a ray from our network through Equation 3, we compute $L_r$ for supervision:

$$L_r = \sum_{\mathbf{r} \in \mathcal{R}} \|C(\mathbf{r}) - \hat{C}(\mathbf{r})\|^2, \tag{4}$$

where $\mathcal{R}$ is the set of sampled rays. $C(\mathbf{r})$ is the ground truth color of ray $\mathbf{r}$ in the training images.

**Auxiliary loss tackling imbalanced optimization.** One problem of training MoE-based networks is that the gating network can favor only a few experts (Shazeer et al., 2017). The optimization and utilization of NeRF experts will thus be imbalanced. This could even cause several experts not trained, and then the whole network converges to a sub-optimal solution and cannot scale well through the training since some of the network capacities are not fully utilized. Following Shazeer et al. (2017); Lepikhin et al. (2021), we use an auxiliary loss $L_a$ to regularize the gating network and balance the utilization of NeRF experts. Specifically, following the differentiable load balancing loss proposed in GShard (Lepikhin et al., 2021), we define our auxiliary loss as follows. Given $n$ NeRF experts and a batch $\mathcal{B}$ with $N$ sample points, let $c_i$ be the number of points dispatched to each expert $E_i$ by Top-1. We first compute the gating values $m_i$ distributed to $E_i$ with $m_i = \sum_{\mathbf{x} \in \mathcal{B}} G(\mathbf{x})_i$. The auxiliary loss $L_a$ can be computed as $L_a = \frac{n}{N^2} \sum_i^n c_i m_i$. This auxiliary loss encourages balanced gating because it is minimized when the dispatching is ideally balanced. Under the balanced gating, $c_i$ and $m_i$ are both expected to be $\frac{N}{n}$. Then $\sum_i^n c_i m_i$ will be $N^2/n$. The loss $L_a$ will be 1.

The total loss $L$ is the weighted sum of the above-mentioned two losses:

$$L = L_r + \lambda L_a \tag{5}$$

where $\lambda$ is the weight for our auxiliary loss. We set $\lambda = 5 \times 10^{-4}$ for all our main results and it is sufficient to balance the utilization of experts.

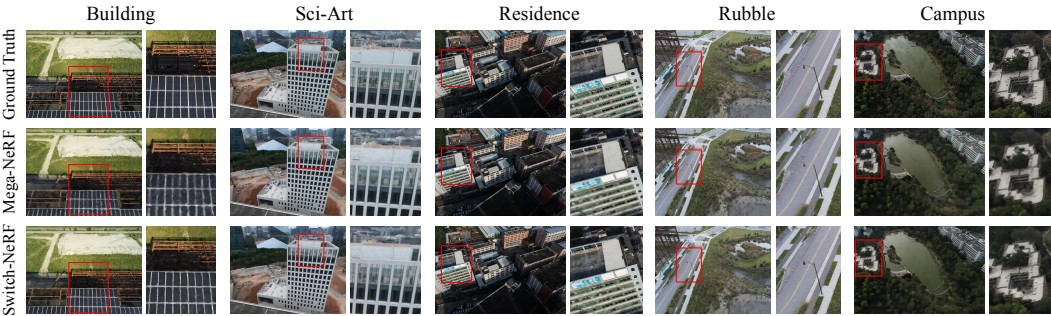

Figure 5: The comparison of the rendered images from Mega-NeRF and our Switch-NeRF. Our method renders more details and tiny structures than Mega-NeRF. Please zoom in to see the details.

Table 1: The testing results of our Switch-NeRF on large-scale datasets. Our method gets state-of-the-art accuracy compared to the dense NeRF, NeRF++ and the sparse Mega-NeRF.

| Dataset | Metrics | NeRF | NeRF++ | SVS | DeepView | Mega-NeRF | Switch-NeRF |
|---------|---------|------|--------|-----|----------|-----------|-------------|
| Building | PSNR↑ | 19.54 | 19.48 | 12.59 | 13.28 | 20.93 | **21.54** |
| | SSIM↑ | 0.525 | 0.520 | 0.299 | 0.295 | 0.547 | **0.579** |
| | LPIPS↓ | 0.512 | 0.514 | 0.778 | 0.751 | 0.504 | **0.474** |
| Rubble | PSNR↑ | 21.14 | 20.90 | 13.97 | 14.47 | 24.06 | **24.31** |
| | SSIM↑ | 0.522 | 0.519 | 0.323 | 0.310 | 0.553 | **0.562** |
| | LPIPS↓ | 0.546 | 0.548 | 0.788 | 0.734 | 0.516 | **0.496** |
| Residence | PSNR↑ | 19.01 | 18.99 | 16.55 | 13.07 | 22.08 | **22.57** |
| | SSIM↑ | 0.593 | 0.586 | 0.388 | 0.313 | 0.628 | **0.654** |
| | LPIPS↓ | 0.488 | 0.493 | 0.704 | 0.767 | 0.489 | **0.457** |
| Sci-Art | PSNR↑ | 20.70 | 20.83 | 15.05 | 12.22 | 25.60 | **26.52** |
| | SSIM↑ | 0.727 | 0.755 | 0.493 | 0.454 | 0.770 | **0.795** |
| | LPIPS↓ | 0.418 | 0.393 | 0.716 | 0.831 | 0.390 | **0.360** |
| Campus | PSNR↑ | 21.83 | 21.81 | 13.45 | 13.77 | 23.42 | **23.62** |
| | SSIM↑ | 0.521 | 0.520 | 0.356 | 0.351 | 0.537 | **0.541** |
| | LPIPS↓ | 0.630 | 0.630 | 0.773 | 0.764 | 0.618 | **0.609** |

## 4 EXPERIMENTS

### 4.1 DATASETS, METRICS AND VISUALIZATION

**Datasets.** We use the Building, Rubble datasets from Mill 19 (Turki et al., 2022) and Residence, Sci-Art, Campus datasets from UrbanScene3D (Liu et al., 2021) to evaluate our Switch-NeRF. Each scene contains thousands of high-resolution images. The camera parameters for all the images are the same as Mega-NeRF (Turki et al., 2022). These datasets cover large enough areas while they can still be handled by consumer-level workstations with commonly used GPUs.

**Metrics.** We use the PSNR, SSIM (Wang et al., 2004) (both higher is better), and VGG implementation of LPIPS (Zhang et al., 2018) (lower is better) to quantitatively evaluate our results on the novel view synthesis. The PSNR is to measure the mean squared error between two images in logarithmic space. The SSIM focuses more on structural similarity. The LPIPS measures perceptual similarity.

**Visualization.** Besides the rendered images, we visualize the 3D radiance fields. We sample 3D points along rays and use the $\alpha = 1 - \exp(-\sigma_i \delta_i)$ as the opacity of each 3D point. We use Point Cloud Library (Rusu & Cousins, 2011) to show the color and opacity of each 3D point.

### 4.2 SETTING

Similar to NeRF++ (Zhang et al., 2020) and Mega-NeRF, the 3D scene space is spilt into a foreground and a background. We use 8 experts with Top-1 gating and $C_f = 1.0$ for the foreground to end-to-end learn scene decomposition, and one original NeRF in the background. This differs from Mega-NeRF using a foreground and a background NeRF for each of its 8 sub-networks, requiring more network parameters. Each of our NeRF experts contains 7 layers with each layer 256 channels. We sample 256 coarse and 512 fine points per ray in the main network and 128/256 samples in the background network. We use 8 NVIDIA RTX 3090 GPUs for distributed data-parallel training and sample 1024 rays for each GPU. We use Adam optimizer (Kingma & Ba, 2015) and a learning rate decaying exponentially from $5 \times 10^{-4}$ to $5 \times 10^{-5}$, and use bfloat16 in training and float16 in testing to reduce memory and time. We train 500k iterations for each dataset and test on validation images.

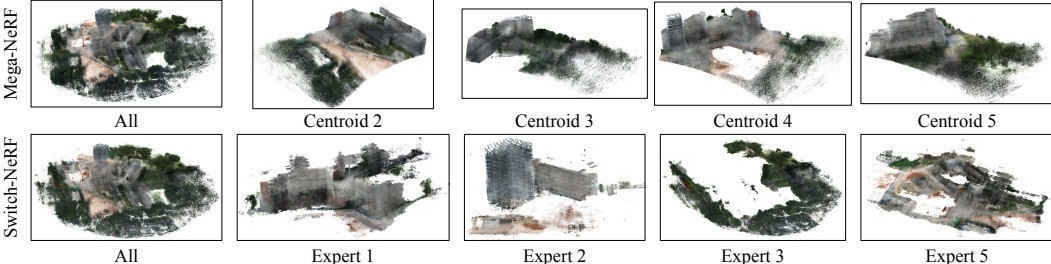

Figure 6: The visualization of 3D radiance fields handled by experts in Switch-NeRF and sub-networks in Mega-NeRF. The Mega-NeRF just regularly decomposes the scene. Our experts are roughly specialized to different semantic parts of the scene such as buildings, trees and grounds.

## 4.3 BENCHMARK PERFORMANCE

The main quantitative results are reported in Table 1 and qualitative results in Fig. 5. The statistics of NeRF (Mildenhall et al., 2020), NeRF++ (Zhang et al., 2020), SVS (Riegler & Koltun), Deep-View (Flynn et al., 2019), and Mega-NeRF are quoted from Mega-NeRF. The MLP width of NeRF and NeRF++ is both set to 2048 to obtain a similar capacity between Mega-NeRF and Switch-NeRF.

As shown in Table 1, our Switch-NeRF achieves state-of-the-art accuracy on all the datasets in terms of PSNR, SSIM, and LPIPS. It produces much better accuracy than NeRF and NeRF++. This confirms the effectiveness of scaling NeRF with sparse Mixture-of-Experts instead of densely expanding the network. Our method also outperforms the sparse Mega-NeRF, showing the advantage of learning decomposition instead of the hand-crafted scene decomposition. Fig. 5 presents the rendered images of Mega-NeRF and Switch-NeRF. Our method can render more tiny structures and details compared to Mega-NeRF. These main results demonstrate the overall performance of our method.

## 4.4 MODEL ANALYSIS

Besides the main results, we perform ablation experiments to analyze different aspects of our model. The Mill 19 Building scene is used by default.

**Decomposition.** We analyze our learning-based decomposition compared to other decomposition methods. The random decomposition randomly dispatches 3D points into sub-networks. The distance decomposition dispatches 3D points based on the 3D distances in training and testing. The network design and the training setup for these variants are the same as Switch-NeRF. We also include the results of Mega-NeRF, which uses distances to cluster pixels

Table 2: The ablation results on decomposition, gating network design, auxiliary loss $L_a$ and the unified head in Switch-NeRF.

|  |  | PSNR↑ | SSIM↑ | LPIPS↓ |
|---|---|---|---|---|
| Decom. | Random | 20.06 | 0.455 | 0.575 |
|  | Distance | 20.50 | 0.547 | 0.499 |
|  | Mega-NeRF | 20.93 | 0.547 | 0.504 |
| Gate | Linear | 20.75 | 0.532 | 0.524 |
|  | w/o Norm | 21.39 | 0.577 | **0.474** |
|  | w/o $L_a$ | 15.32 | 0.363 | 0.769 |
|  | w/o unified head | 19.72 | 0.472 | 0.561 |
|  | Switch-NeRF | **21.54** | **0.579** | **0.474** |

in training and 3D points in testing. The results are shown in Table 2. The random decomposition cannot fully utilize more parameters in the network. The distance decomposition performs better and is similar to Mega-NeRF. Our Switch-NeRF with learned decomposition achieves the best accuracy.

**NeRF expert specialization.** We visualize the 3D radiance fields of the UrbanScene3D-Sci-Art for Mega-NeRF and Switch-NeRF in Fig. 6. The Mega-NeRF partitions the scene into regular parts. In Switch-NeRF, the experts have been roughly specialized to different semantic parts of the scene. Experts 1 and 2 focus on different buildings. Expert 3 focuses on the green grasses and trees on the ground. Expert 5 specializes on other parts of the ground. Our Switch-NeRF can achieve more reasonable and semantically meaningful scene decomposition in a learning way.

Table 3: The accuracy, test time, test memory and parameters numbers of Switch-NeRF and Mega-NeRF with different expert or sub-network numbers. From 8 to 16 experts, our network scales much better than Mega-NeRF.

| Model | PSNR↑ | SSIM↑ | LPIPS↓ | Mem.↓ | Time↓ | Param.↓ |
|---|---|---|---|---|---|---|
| Switch-4 | 21.00 | 0.547 | 0.504 | 5825M | 106s | 2.78M |
| Mega-8 | 20.93 | 0.547 | 0.504 | 6935M | 87.9s | 10.8M |
| Switch-8 | 21.54 | 0.579 | 0.474 | 5847M | 110s | 4.53M |
| Mega-16 | 21.47 | 0.590 | 0.462 | 8042M | 101s | 21.6M |
| Switch-16 | **22.49** | **0.625** | **0.429** | 5876M | 118s | 8.05M |

Table 4: The accuracy, training memory and training time with Top-1, Top-2 and capacity factors. Increasing $C_f$ or using Top-2 in training can improve the accuracy while increasing the training time and memory.

| Top-$k$ | Capacity | PSNR↑ | SSIM↑ | LPIPS↓ | Mem.↓ | Time↓ |
|---|---|---|---|---|---|---|
| 1 | 1.0 | 21.54 | 0.579 | 0.474 | **10182M** | **42.5h** |
| 1 | 1.5 | 21.70 | **0.594** | **0.463** | 12271M | 47.4h |
| 2 | 1.0 | **21.77** | 0.590 | 0.465 | 14548M | 58.7h |

Table 5: Comparisons of parameter number, time, memory and FLOPs. Although with more training time and memory, our network uses less testing memory and has much less parameters while achieving a better accuracy.

| Method | Test | | Train | | Param.↓ | FLOPs↓ |
| | Mem.↓ | Time↓ | Mem.↓ | Time↓ | | |
|---|---|---|---|---|---|---|
| Mega-NeRF | 6935M | 87.9s | 5124M | 30.7h | 10.8M | 0.79M |
| Switch-NeRF | 5847M | 110s | 10182M | 42.5h | 4.53M | 0.99M |

Table 6: The accuracy and testing time of uniform dispatch with $C_f = 2.0$ and $4.0$ and full dispatch. The full dispatch clearly perform better with less time usage.

| Dispatch | PSNR↑ | SSIM↑ | LPIPS↓ | Time (s)↓ |
|---|---|---|---|---|
| Uniform 2.0 | 17.82 | 0.410 | 0.563 | 131 |
| Uniform 4.0 | 20.07 | 0.521 | 0.507 | 182 |
| Full | **21.54** | **0.579** | **0.474** | **110** |

**Gating network.** We analyze different designs of our gating network. As shown in Table 2, the Linear gating just feeds the PE($\mathbf{x}$) into a trainable linear layer. The w/o Norm version does not have a LayerNorm. We also evaluate the gating without the auxiliary loss $L_a$. The Linear gating with fewer parameters does not generate satisfactory results. The MLP+Norm can boost the performance of our whole network. The network without auxiliary loss $L_a$ does not even converge. This shows $L_a$ is vital for the success of MoE methods, consistent with the observations in Shazeer et al. (2017).

**Unified head.** We remove the shared unified head and add a head separately for each NeRF expert. Table 2 shows that without the unified head the accuracy significantly decreases. We also observe that in this case, the network is not robust enough to the mixed-precision training and testing. A possible reason is that the gating value is directly multiplied to the predictions rather than the high-level features, which makes the gating and prediction unstable in training and testing. This verifies our motivation to design a unified head for multiple NeRF experts.

**Scalability.** Table 3 shows the accuracy and efficiency of Switch-NeRF with different numbers of experts. With only 4 experts, we already achieve similar accuracy to Mega-NeRF using 8 sub-networks. Compared to the model with 8 experts, the model with 16 experts increases the performance remarkably, while without a large increase of the memory and time in testing. Compared to Mega-NeRF with 16 sub-networks, our Switch-NeRF with 16 experts scales much better with a higher accuracy, fewer number of the parameter and less memory footprint in testing. All of these prove the scalability of our Switch-NeRF. It can obtain much better results by increasing network capacity while maintaining almost constant computational and memory costs.

**Effect of top-2 and capacity factor.** Table 4 studies using Top-2 and $C_f$ when training Switch-NeRF. Increasing $C_f$ from 1.0 to 1.5 or using Top-2 in training can improve the accuracy while increasing the training time and memory, especially for Top-2. With our design of Switch-NeRF, Top-1 with $C_f = 1.0$ already obtains good results with acceptable efficiency, suggesting that increasing $C_f$ instead of $k$ is better because $C_f$ in training does not affect testing with the full dispatch.

**Efficiency.** In Table 5, we compare the efficiency of Switch-NeRF with Mega-NeRF. Our model achieves better accuracy using only *half* of the parameters of Mega-NeRF. Mega-NeRF uses a background NeRF and different versions of appearance embeddings (AE) for each sub-network, leading to much more parameters. As Switch-NeRF is trained end-to-end, it only uses one background NeRF and one version of AE. Our network is more parameter efficient and with better accuracy. Since it has a gating network and is trained end-to-end, it reasonably requires more floating point operations (FLOPs) for each point and costs slightly more memory and time for training. Notably, compared to Mega-NeRF, it uses around 20% less testing memory, and only a very minor increase in testing time. Our Switch-NeRF achieves a good efficiency together with a better accuracy.

**Full dispatch.** We test our trained model with a uniform dispatch and our full dispatch discussed in Sec. 3.3. A capacity factor $C_f$ of 2.0 and 4.0 with Batch Prioritized Routing (Riquelme et al., 2021) is used in the uniform dispatch. Table 6 shows that the results of the uniform dispatch with $C_f = 2.0$ largely decreases. Although the results with $C_f = 4.0$ improve, they are still far lower than the full dispatch. Moreover, it costs more time than the full dispatch because of a large $C_f$ with zero padding. Fig. 4 shows that images rendered with the uniform dispatch contain much more artifacts than the full dispatch, proving the effectiveness of the full dispatch in Switch-NeRF.

## 5 CONCLUSION

In this paper, we present Switch-NeRF. To the best of our knowledge, it is the first sparse large-scale NeRF with learnable scene decomposition. We propose an end-to-end mixture-of-NeRF-experts framework to learn scene decomposition jointly with NeRF. We further design and implement an efficient network architecture and a full dispatch strategy to boost accuracy. Extensive experiments demonstrate that our network can learn more reasonable scene decomposition and shows state-of-the-art accuracy on the scene synthesis compared to hand-crafted decomposition methods.

ACKNOWLEDGEMENTS

This research is supported in part by HKUST-SAIL joint research funding, the Early Career Scheme of the Research Grants Council (RGC) of the Hong Kong SAR under grant No. 26202321 and HKUST Startup Fund No. R9253.

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

APPENDIX

## A    BUNGEENERF DATASET

We compare Switch-NeRF with BungeeNeRF (Xiangli et al., 2022) on the 56Leonard and Transamerica datasets provided from BungeeNeRF. We first analyze the different motivations and focuses between the BungeeNeRF and our Switch-NeRF. The BungeeNeRF focuses on modeling scenes with multi-scales and different levels of details, while Switch-NeRF targets large-scale scene modeling with large data volumes. BungeeNeRF has several special designs for training. It decomposes the training data by predefined scales and uses multi-level supervision for its middle blocks. It designs special data feeding scheme and model growing training. We conducted experiments on the BungeeNeRF dataset, and show the performance comparison in Table 7. It is worth noting that the data volume of BungeeNeRF (around 450 images with 640x360 resolution for each scene) is similar to the vanilla NeRF but is significantly smaller compared to that of Mega-NeRF (the Building dataset with 1940 images of 4608x3456 resolution). Thus, we use less experts for BungeeNeRF dataset. As seen in Table 7, our Switch-NeRF with 4 experts can outperform the BungeeNeRF in all scales.

Table 7: Comparisons of PSNR on the 56Leonard and Transamerica scenes in BungeeNeRF. Switch-NeRF with 4 experts outperforms BungeeNeRF in all scales.

| Scene | Method | Scale 1 | Scale 2 | Scale 3 | Scale 4 | Avg. |
|---|---|---|---|---|---|---|
| 56Leonard | BungeeNeRF | 24.120 | 24.345 | 25.382 | 25.112 | 24.513 |
| | Switch-NeRF-2 | 23.666 | 24.329 | 24.676 | 24.475 | 24.162 |
| | Switch-NeRF-4 | **24.624** | **25.263** | **25.844** | **25.697** | **25.196** |
| Transamerica | BungeeNeRF | 24.608 | 24.350 | 24.357 | 24.608 | 24.415 |
| | Switch-NeRF-2 | 24.468 | 24.233 | 24.410 | 23.495 | 24.220 |
| | Switch-NeRF-4 | **25.242** | **25.276** | **25.504** | **24.672** | **25.226** |

## B    BLOCK-NERF DATASET

We train a Switch-NeRF on the San Francisco Mission Bay Dataset of Block-NeRF (Tancik et al., 2022). It consists of 12,000 images with an image resolution of around 1200x900. Although the data volume is similar to the datasets used by Mega-NeRF and Switch-NeRF, the training setting of Block-NeRF is quite different from ours. The Block-NeRF uses much larger computing and memory resources in its training setting. It trains each scene block with 32 TPU v3 cores combining offer 512 GB memory, which is not easy to be followed by the common computer workstations. The batch-size of Block-NeRF for each block is 16384, significantly larger than the averaging batch-size 1024 for each sub-network in Mega-NeRF and Switch-NeRF. The networks of BlockNeRF are trained with full precision while ours with bfloat16 precision. Besides, an important part of the training data, i.e., the masks for dynamic objects in the scenes used by Block-NeRF, are also not released in the available training data. The Block-NeRF does not open-source its code, which makes us difficult to align its hyper-parameters on their training data.

Due to our lack of comparable computing resources and the unavailability of the same training data used in Block-NeRF, it is not feasible for us to train a Switch-NeRF based on the same setting as Block-NeRF, to allow a direct and fair comparison. We thus train our network on the Block-NeRF dataset with a similar setting used in our paper. We use 8 NVIDIA RTX 3090 GPUs to train a Switch-NeRF model with 8 experts, and sample 1664 rays for each GPU, resulting in a batch-size of 1664 on average for each expert. This is far less than the batch-size of 16384 for each Block in Block-NeRF. The width of each layer is set as 512 as in Block-NeRF. We also use the positional encoding of Mip-NeRF as used by Block-NeRF. We sample 256 points each ray for the coarse and fine networks in training, and 512 points each ray for testing. We use the validation dataset of Block-NeRF to evaluate the performance with the PSNR, SSIM and LPIPS metrics. The experimental results are shown in Table 8.

Table 8: Accuracy of Switch-NeRF on the San Francisco Mission Bay Dataset proposed by Block-NeRF. We train the Switch-NeRF with limited computing resources, , half float precision, and much smaller batch-size compared to Block-NeRF. Besides, Switch-NeRF does not utilize the important dynamic object masks that are used by Block-NeRF, as they are not available in the released training dataset.

| Method | PSNR↑ | SSIM↑ | LPIPS↓ |
|---|---|---|---|
| Switch-NeRF (8 experts) | 23.86 | 0.762 | 0.489 |

## C  TWO GATING OPERATIONS

We add an additional ablation study on two gating operations. We train a Switch-NeRF with 2 gating networks, each acting as a gating operation. The first gating network is added after the 1-st linear layer of the main network, and the second gating network is added after the 4-th linear layer of the main network. The total number of expert layers is the same as that of the Switch-NeRF using 1 gating network to make them have similar network capacity. The results in terms of both accuracy and efficiency are shown in Table 9. From the results, we can clearly see that two gating operations cost more training time, training memory, and testing time, while achieving a similar accuracy to the model with one gating operation. This suggests that the number of gating operations is not critical for the model performance, and one gating operation already shows sufficient capabilities in the dispatch of points.

Table 9: Accuracy and efficiency with different numbers of gating operations. The Switch-NeRF with two gating operations and the same number of expert layers, does not improve the rendering accuracy over the model with one gating operation, while consuming more memory and time for training and testing.

| Gating number | PSNR↑ | SSIM↑ | LPIPS↓ | Train Mem. | Train Time | Test Mem. | Test Time |
|---|---|---|---|---|---|---|---|
| 1 | 21.54 | 0.579 | 0.474 | 10182M | 42.5h | 5847M | 110s |
| 2 | 21.55 | 0.574 | 0.477 | 15315M | 56.2h | 5838M | 152s |

## D  MORE EXPERTS

The expert number directly controls the network capacity of Switch-NeRF. With large-scale scenes we may need to increase the expert number to get better results. In this section we add more experiments on the expert number. In Table 10, the results show that more experts can consistently produce better results for larger-scale dataset.

We additionally visualize the 3D radiance fields of a Switch-NeRF trained with 16 experts on the UrbanScene3D-Sci-Art dataset in Figure 7. Compared with the Figure 6 in the main paper, we can see that our network can still roughly specialized to different fine-grained semantic parts of the scene.

Table 10: The accuracy of different expert numbers. Increasing expert number can consistently improve the accuracy.

| Method | Building PSNR↑ | Building SSIM↑ | Building LPIPS↓ | Sci-Art PSNR↑ | Sci-Art SSIM↑ | Sci-Art LPIPS↓ | Campus PSNR↑ | Campus SSIM↑ | Campus LPIPS↓ |
|---|---|---|---|---|---|---|---|---|---|
| Switch-NeRF-8 | 21.54 | 0.579 | 0.474 | 26.52 | 0.795 | 0.360 | 23.62 | 0.541 | 0.609 |
| Switch-NeRF-16 | **22.49** | **0.625** | **0.429** | **27.17** | **0.819** | **0.327** | **24.40** | **0.578** | **0.553** |

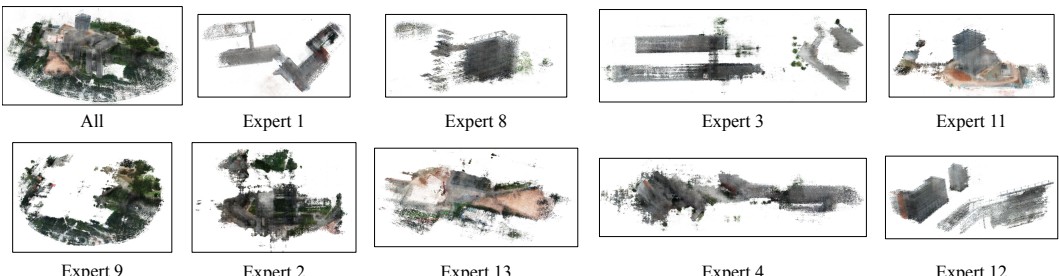

Figure 7: The visualization of 3D radiance fields handled by experts in Switch-NeRF with 16 experts on the UrbanScene3D-Sci-Art dataset.

# E    FAST RENDERING

We explore the capability of Switch-NeRF to be integrated into fast rendering techniques based on octrees similar to the interactive exploration used in Mega-NeRF-Dynamic Turki et al. (2022). As the detailed evaluation protocol for the faster rendering is not provided by Mega-NeRF-Dynamic, we thus define a protocol to evaluate for both methods, on the validation images of each dataset. Specifically, we first convert the trained model into a coarse octree. Given a view point, we render a coarse image directly from the coarse octree. Then, we follow the dynamic octree refinement strategy proposed by Mega-NeRF-Dynamic, to refine the octree for the current view point for several rounds by querying the trained model. Here, we consider 16 rounds. We finally render a refined image from the refined octree. We evaluate both Mega-NeRF model and Switch-NeRF model on the same fast-rendering protocol we just described, and report the PSNR and average refinement time of octree in Table 11, and images in Figure 8. As shown in Table 11, since the octrees before refinement are very coarse, the coarse images for both Mega-NeRF and Switch-NeRF are of low quality. After the dynamic octree refinement, the quality of images improves. Since the resolution of refined octrees is still limited, the quality is not comparable to the original model. However, our Switch-NeRF can obtain similar or better results on the validation datasets compared to Mega-NeRF. It should be noted that the octrees do not handle the background, and thus the rendered images may have black regions in both Mega-NeRF and Switch-NeRF, which decreases the PSNR values. However, as shown in Figure 8, our method can render good quality of refined images in the foreground. This shows our Switch-NeRF can be flexibly and effectively integrated into existing faster rendering techniques.

Table 11: The PSNRs of images rendered from the coarse octrees and refined octrees of Switch-NeRF and Mega-NeRF, and the average time used for dynamic octree refinement. Switch-NeRF can get similar or better results on the validation datasets compared to Mega-NeRF. Note that the octrees do not handles background so the rendered images may have black regions in both Mega-NeRF and Switch-NeRF, which will decrease the PSNR values.

|  |  | Building | Rubble | Residence | Sci-Art | Campus | Time |
|---|---|---|---|---|---|---|---|
| Mega-NeRF | Coarse | 14.66 | 14.99 | 12.15 | 10.86 | 14.95 | |
|  | Refined | **16.30** | 16.93 | 13.85 | 13.50 | 18.30 | 6.3s |
| Switch-NeRF | Coarse | 14.29 | 15.16 | 12.05 | 10.74 | 16.05 | |
|  | Refined | 16.29 | **17.04** | **13.96** | **13.61** | **18.66** | 13.0s |

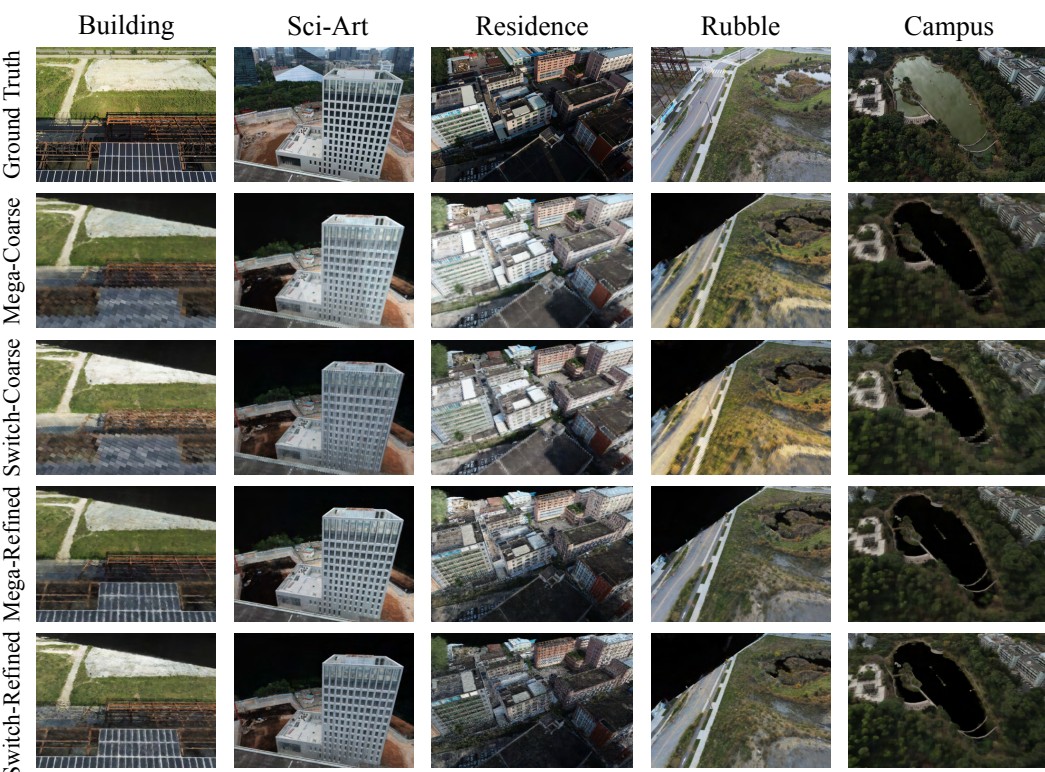

Figure 8: Images of Switch-NeRF and Mega-NeRF rendered by coarse octrees and refined octrees. Note that the octrees do not handles background so the rendered images may have black regions in both Mega-NeRF and Switch-NeRF. This will decrease the PSNR values. However, from the visualization, we can see that Our method can render good quality of refined images in the foreground.

