# OpenReview forum: "Switch-NeRF: Learning Scene Decomposition with Mixture of Experts for Large-scale Neural Radiance Fields"
_ICLR.cc/2023/Conference — ICLR 2023 poster_

### Official Review · Reviewer_YY1K · 2022-10-24

**Confidence:** 3
**Correctness:** 4
**Technical Novelty And Significance:** 3
**Empirical Novelty And Significance:** 3
**Recommendation:** 8

**Clarity, Quality, Novelty And Reproducibility:**

Overall, the paper is detailed and clear. The main novel contribution is the usage of a Sparsely Gated MoE for large-scale NeRF training. Given that sparse MoEs are not common for this or similar uses, this contribution is novel, and a significant amount of exploration is done to get good performance out of the approach. The hyperparameters and implementation are described in detail for reproducibility.

**Strength And Weaknesses:**

Strengths:
- Demonstrates good results, improving on rendering quality relative to Mega-NeRF while keeping parameters and FLOPs roughly constant.
- Motivates the approach well and visualizes the result of the gating network by showing that different experts learn to handle different parts of the scene.
- Includes a good about of ablations and comparisons to quantitatively evaluate the impact of their choices.

Weaknesses:
- (Minor) The use of the composition operator to describe the series of linear layers, softmaxes, ReLUs, etc, is non-conventional. Could be presented more clearly with a visual depiction or simply function application F(G(x)).
- The presentation is general in the k (for Top-k) but in practice k=1. When k=1, it seems confusing why you would multiply the network output by G(x), since that's just a linear scaling factor. It also seems unclear why this would help the gating network train, although it is true that it makes it differentiable with respect to the output. If k > 1, then the gradient for a sample would favor the experts which provide a better input for that sample, but with k=1 this is less intuitive. I think the clarity of the paper could be improved either by elucidating this point, or perhaps by just rewriting this with k fixed to 1 instead of general.
- (Minor) The authors observe that "we observe that dropping sample points can significantly decrease the test accuracy." It is unclear to me what happens when a sample is dropped. Is the 2nd ranked expert used? Or is the value fed to the uniform head simply Linear(x)? In either case,  it seems inappropriate to drop samples, but would be helpful to understand what the alternative is.
- (Minor) Missing parentheses in "A capacity factor Cf of 2.0 and 4.0 with Batch Prioritized Routing Riquelme et al"

**Summary Of The Paper:**

The paper proposes to use a sparsely gated mixture of experts (MoE) model for NeRF models trained on large-scale scenes. The paper evaluates a variety of design choices and hyperparameters and compares results to Mega-NeRF, which also uses multiple NeRFs to model a large scene but simply uses the distance to a centroid to determine which NeRF to use.

**Summary Of The Review:**

This paper does one thing – sparse MoEs for large-scale NeRFs – and does it well, exploring many different choices and ablations and showing what is needed to get this approach to work. The approach is motivated well in the discussion and by the results. In summary, this is a strong contribution to the growing field of large-scale NeRF modeling.

---

> ### Author Response · Authors · 2022-11-19
> **Response to Reviewer YY1K**
>
> We thank the reviewer for the positive feedback and insightful comments. We address the detailed questions and comments below.
>
> >R4.Q1: Composition operator.
>
> Thank the reviewer for the advice on the illustration of the gating network. We already depict the structure of the gating network in Figure 3 in the original submission. We modified the equation according the suggestion in the revised paper.
>
> >R4.Q2: Rewriting with k fixed to 1 instead of general.
>
> Thank the reviewer for the advice on the description of our gating network. We have fixed k to 1 in the description in Sec 3.1 and have rewritten the equation to eliminate ambiguity. The k=1 setting was first proposed by Switch Transformer. Although it is less intuitive, its effectiveness and efficiency have been analysed and proved by Switch Transformer. It is also a very interesting problem to investigate the mechanisms behind k=1.
>
> >R4.Q3: What happens when a sample is dropped?
>
> When a sample is dropped, this sample is not processed by any expert. The expert output for this sample is set as zeros. This happens when a capacity factor is set in the training to obtain uniform tensor shapes and balanced computation. In the testing, we use the proposed full dispatch strategy described in our paper, which enables no dropping of samples.
>
> >R4.Q4: Missing parentheses.
>
> Thank the reviewer for the careful reminder. We have fixed the parentheses in the revised paper.

---

### Official Review · Reviewer_7KWF · 2022-10-24

**Confidence:** 4
**Correctness:** 3
**Technical Novelty And Significance:** 3
**Empirical Novelty And Significance:** 3
**Recommendation:** 6

**Clarity, Quality, Novelty And Reproducibility:**

The quality, clarity, and originality is on of above the average of ICLR accepted papers.

The reproducibility will be sounder if the authors can provide the code of it because optimizing a gating function together with NeRF model may be tricky thus checking the whole training process including all hyper-parameters may be important to determine the reproducibility.

**Strength And Weaknesses:**

# Strength
> + Target an interesting problem: while most large-scale NeRF use heuristic hand-crafted scene decomposition, how to make the decomposition learnable and different for different datasets is an interesting direction to explore.
> + Comprehensive experiments: The experiments, especially ablation studies are sufficient to support the proposed Switch-NeRF is effective to boost the PSNR of NeRF on large-scale scenes.
> + Well written paper: the paper is well written and easy to follow.

# Weaknesses
> + Expandability to new environment: One advantage of NeRF with hand-crafted scene decomposition is that the independence of different blocks or decomposed areas are explicit, which enables the ability to expand the environment with additional NeRF models or update blocks without retraining the entire environment. It is unclear how the proposed Switch-NeRF can be expanded to new environment without retraining the entire environment because the decomposition assignment to previous environment maybe changed when new environment is involved in.
> + Ability of "baking" for fast rendering: Mega-NeRF has shown its ability to be converted to Plenoctree or KiloNeRF-based models for interactive rendering, thus it may not be fair to compare with  Mega-NeRF  without such conversion in Table 3. Also, because the proposed Switch-NeRF has a gating function for the decomposition assignment and such assignment is implicit, it is not clear how Switch-NeRF can be converted to other representations for fast rendering like Mega-NeRF. If it can, not sure whether it will suffer from larger PSNR drop as compare to Mega-NeRF with explicit decomposition. Since speeding the rendering to real-time is an important feature for interactive view as mentioned in Mega-NeRF, it would be better if the authors can add more discussion on it.
> + Need more analysis on the efficiency: as claimed by the authors, "As Switch-NeRF is trained end-to-end, it only uses one background NeRF and one version of AE.", the training cost  should not be larger than baselines, e.g., Mega-NeRF. However, it cost more than Mega-NeRF (42 vs. 30 h) as shown in Table 5. The corresponding explanation, "Since it has a gating network and is trained end-to-end, it is reasonable that it requires more floating point operations (FLOPs) for each point and costs slightly more memory and time for training", is not clear enough. What is main cause of more training cost? More difficult to optimize because of the gating function or larger models to forward in each iteration?

**Summary Of The Paper:**

This work presents a framework for joint learning scene decomposition and NeRF, targeting NeRF for large-scale scene. It is well-recognized that decomposing NeRF is needed for large-scale scene, i.e., a standalone NeRF model for a specific region, because of the model can be extremely large without decomposition and thus cannot be optimized with existing solutions. However, most large-scale NeRF use heuristic hand-crafted scene decomposition, i.e., decomposing into different blocks or balls in the 3D space, which maybe sub-optimal. Thus, this work propose to learn such decomposition configurations together to the NeRF model. Specifically, it inserts a gating function before querying each points' embeddings in MLP and such a gating function will assign points to different NeRF model experts. The experiments show that such a learnt decomposition can achieve better PSNR than heuristic hand-crafted  solutions.

**Summary Of The Review:**

Overall, this paper provides some interesting insights to the community. It is nice to see learnable decomposition can surpass heuristic hand-crafted scene decomposition although it should be true intuitively.
However, I do have some concerns on its efficiency, expandability to new environment, and ability of "baking" for fast rendering. I will raise my scores if the authors can solve these concerns.

---

> ### Author Response · Authors · 2022-11-19
> **Response to Reviewer 7KWF (1/2)**
>
> We thank the reviewer for the positive feedback and insightful comments. We address the detailed questions and comments below.
>
> >R3.Q1: Expandability to new environments.
>
> Expandability of NeRF to new environments is a very interesting problem. We analyze two cases to show the possibility of our Switch-NeRF to be expanded to new environments. The first case is that a part of the scene is changed. It is possible to fine-tune Switch-NeRF to quickly adapt to the changed part. We can fix the gating network and most of the expert parameters, and only tune several layers in experts and heads. The fine-tuning techniques are widely explored in MoE methods in the NLP domain, such as Switch Transformer. The second case is that a new environment is involved in the modeling. In this case, since the new environment is not tightly related to the old environment, we can train another gating network and a set of experts on this new environment. Then, we can combine the new model with the old model by only fine-tuning the fusion head.
>
> >R3.Q2: Fast rendering.
>
> Thank the reviewer for the question. Our network uses an MLP to dispatch 3D points. Mega-NeRF uses distance to dispatch 3D points in testing. Therefore, it is straightforward to integrate our network into the fast rendering pipeline of Mega-NeRF, based on octrees similar to the interactive exploration used in Mega-NeRF-Dynamic. As the detailed evaluation protocol for the fast rendering is not provided by Mega-NeRF-Dynamic, we thus define a protocol to evaluate for both methods, on the validation images of each dataset. Specifically, we first convert the trained model into a coarse octree. Given a view point, we render an coarse image directly from the coarse octree. Then, we follow the dynamic octree refinement strategy proposed by Mega-NeRF-Dynamic, to refine the octree for the current view point for several rounds by querying the trained model. Here, we consider 16 rounds, while in the paper of Mega-NeRF-Dynamic, it only mentions several rounds without giving a concrete number of rounds for the refinement. We finally render a refined image from the refined octree. We do not use the guided sampling function in the fast-rendering codes of the Mega-NeRF-Dynamic, because we cannot successfully run this function with both the official Mega-NeRF model and our Switch-NeRF model. The authors also mentioned on GitHub that "This is a preliminary release and there may still be outstanding bugs" for the fast-rendering code. We also approached the authors about the code issue but we received no response. Therefore, we finally evaluate both Mega-NeRF model and Switch-NeRF model on the same fast-rendering protocol we just described, and report the PSNR metric and average octree refinement time in the table below, and the Table 11 and rendered images in Figure 8 in the Appendix E. As shown in the table below, since the octrees before refinement are very coarse, the coarse images for both Mega-NeRF and Switch-NeRF are of low quality. After the dynamic octree refinement, the quality of images improves. Since the resolution of the refined octree is still limited, the quality is not comparable to the original model. However, our Switch-NeRF can obtain similar or better results on the validation datasets compared to Mega-NeRF. It should be noted that the octrees do not handle the background, and thus the rendered images may have black regions for both Mega-NeRF and Switch-NeRF models, which decreases the PSNR values. However, as shown in Figure 8, our method can render the foreground with a good quality in the refined images. These results suggest that our Switch-NeRF can be flexibly and effectively integrated into existing fast rendering techniques. Please see Figure 8 in the Appendix E for the visualization results.
>
> Table: The PSNRs of images rendered from the coarse octrees and refined octrees of Switch-NeRF and Mega-NeRF, and the average time used for dynamic octree refinement. Switch-NeRF can get similar or better results on the validation datasets compared to Mega-NeRF. Note that the octrees do not handles background so the rendered images may have black regions in both Mega-NeRF and Switch-NeRF, which will decrease the PSNR values.
>
> | | | Building    | Rubble  | Residence | Sci-Art | Campus | Time  |
> |-------------|---------|-----------|---------|--------|-------|-------|-------|
> | Mega-NeRF   | Coarse  | 14.66     | 14.99   | 12.15  | 10.86 | 14.95 |       |
> |             | Refined | **16.30**     | 16.93   | 13.85  | 13.50 | 18.30 | 6.3s  |
> | Switch-NeRF | Coarse  | 14.29     | 15.16   | 12.05  | 10.74 | 16.05 |       |
> |             | Refined | 16.29     | **17.04**   | **13.96**  | **13.61** | **18.66** | 13.0s |

---

> > ### Author Response · Authors · 2022-11-19
> > **Response to Reviewer 7KWF (2/2)**
> >
> > >R3.Q3: Efficiency analysis.
> >
> > The main reason of a higher training cost is due to the additional computation of the gating operation. In the experiments, as shown in the paper, we train the same number of iterations for each scene as the Mega-NeRF, so the time delay is not caused by the optimization iterations. We apologize that we forgot to show the number of training iterations in Sec. 4.2. We add it in the revised paper. We also add an additional ablation study on the number of gating operations to analyse the efficiency, as shown in Table 9 in Appendix C. The results in Table 9 show that, with 2 gating operations and without changing the network depth, the training time increases from 42.5h to 56.2h, and the test time increases from 110s to 152s. This indicates that the gating operations causes the main time delay in training. For the back-ground NeRF, although we use only one background NeRF, each background sample still needs to be processed by this NeRF, so the computation is not reduced. However, as the background data is only a small portion of the whole dataset, the background NeRF does not largely affect the efficiency.
> >
> > >R3.Q4: Reproducibility.
> >
> > We will release codes, trained models, and hyper parameters of all the experiments upon acceptance, to benefit the research community of the large-scale scene modeling with NeRF.

---

### Official Review · Reviewer_QZyS · 2022-10-25

**Confidence:** 4
**Correctness:** 3
**Technical Novelty And Significance:** 3
**Empirical Novelty And Significance:** 3
**Recommendation:** 6

**Clarity, Quality, Novelty And Reproducibility:**

Please refer to the above for clarity, quality, and novelty. For reproducibility, I think a graduate student can reproduce it.



**Strength And Weaknesses:**

Strength:
- Overall, I appreciate the idea of applying a MOE network to large-scale neural rendering. Some detailed designs in this paper make it to be not so trivial (Not a simple "A+B" approach)
- The paper shows its approach in detail, and the structure is clear.
- The experiments are tested on different datasets, and the ablation studies are rich.


Weaknesses:
Although I am interested in the claims made by the authors, I have some questions and concerns. It will be better if the authors can provide more insights to the claims and experiments.

- For existing methods, “the different sub-networks are typically trained separately”. How do the expert networks in this paper connect with each other? The separate training in Block-NeRF helps it achieve fine-grained results at each block. Does Switch-NeRF achieve better results than Block-NeRF. What are the benefits of training together? Or how does one expert affect the other?

- It will be better if the paper can show some comparison and analysis with Block-NeRF and BungeeNeRF/CityNeRF[1]. The evaluation datasets Mill 19 and UrbanScene3D may not well prove the effectiveness of this paper in the large-scale setting. It might be better if the paper can provide some results on the same setting of Block-NeRF and BungeeNeRF/CityNeRF, e.g., Google Map/Earth.

- The qualitative results are not convincing enough. It would be better if the authors could provide a supplementary video.

- As shown in Figure 6, how do different experts in Switch-NeRF learn different semantic parts? What will different experts learn if Switch-NeRF deals with building-level cases? It is also interesting to see how different numbers of experts behave and affect performance.

- Is there any failure case of Switch-NeRF? From Table 1, the improvement on campus/rubble is lower than sci-art/building. How to understand this? It is meaningful to see how Switch-NeRF behaves on different scales, which can help to inspire the community.

- The paper designs “a deeper gating network to guarantee enough parameters to learn robust scene decomposition”. How robust is it?

- Although the paper claims that “Multiple gating operations will remarkably influence training and testing speed.” It would be meaningful if the authors could provide some insights on what will happen if there is two gating operations. How slow will it be, and how will the performance change?


Reference:
[1] Xiangli, Yuanbo, et al. "BungeeNeRF: Progressive neural radiance field for extreme multi-scale scene rendering." The European Conference on Computer Vision (ECCV). Vol. 2. 2022, former version CityNeRF: Building NeRF at City Scale

**Summary Of The Paper:**

This paper studies large-scale neural rendering. To tackle the challenges of lacking universal scene decomposition, not learnable decomposition procedure, and independent sub-networks optimization, the paper proposes Switch-NeRF, a new end-to-end large-scale NeRF with learning-based scene decomposition. It designs a Sparsely Gated Mixture of Experts network to dispatch 3D points to different NeRF sub-networks. The gating network can be optimized together with the NeRF sub-networks for different scene partitions. The method achieves state-of-the-art performances on several large-scale datasets Mill 19 and UrbanScene3D.

**Summary Of The Review:**

Considering the novelty, quality and some concerns of this paper. I vote for borderline now.

---

> ### Author Response · Authors · 2022-11-19
> **Response to Reviewer QZyS (1/3)**
>
> We thank the reviewer for the positive feedback and insightful comments. We address the detailed questions and comments below.
>
> >R2.Q1: How do the expert networks in this paper connect with each other? What are the benefits of training together?
>
> The experts are trained together via sharing the same decision gating network and a unified prediction head network. The experts can affect each other by these two shared networks (i.e., the gating network, and the unified prediction head network) during backward computation in training. Switch-NeRF can jointly learn smooth scene decomposition and consistent scene composition in an end-to-end manner. It does not require any hand-crafted scene-specific decomposition rules as in existing works, and can achieve learnable fusion of the scene representations from different experts. The discussion of Block-NeRF is presented later together with another question.
>
> >R2.Q2: Results on BungeeNeRF dataset.
>
> We add additional experimental results on BungeeNeRF dataset in Appendix A and add a reference of the paper in the related work. We first analyze the different motivations and focuses between the BungeeNeRF and our Switch-NeRF. The BungeeNeRF focuses on modeling scenes with multi-scales and different levels of details, while our Switch-NeRF targets large-scale scene modeling with large data volumes. BungeeNeRF uses several special designs for training, such as decomposing the training data by pre-defined scales and adding multi-level supervision for its middle blocks. We conduct experiments on the BungeeNeRF dataset, and show the performance comparison in the table below and also in Table 7 in Appendix A. It is worth noting that the data volume of BungeeNeRF (around 450 images with 640x360 resolution for each scene) is similar to the vanilla NeRF, but is significantly smaller compared to that of Mega-NeRF (e.g., the Building dataset with 1940 images of 4608x3456 resolution). Thus, we use less experts for BungeeNeRF. As shown in the table below, our Switch-NeRF with 4 experts can outperform the BungeeNeRF at all scales.
>
>
> Table: Comparisons of PSNR on the 56Leonard and Transamerica scenes in BungeeNeRF. Switch-NeRF with 4 experts outperforms BungeeNeRF at all scales.
>
> | **Scene**                        | **Method**    | **Scale 1**     | **Scale 2**     | **Scale 3**     | **Scale 4**     | **Avg.**        |
> |----------------------------------|---------------|-----------------|-----------------|-----------------|-----------------|-----------------|
> | 56Leonard    | BungeeNeRF    | 24.120          | 24.345          | 25.382          | 25.112          | 24.513          |
> |                                  | Switch-NeRF-2 | 23.666          | 24.329          | 24.676          | 24.475          | 24.162          |
> |                                  | Switch-NeRF-4 | **24.624** | **25.263** | **25.844** | **25.697** | **25.196** |
> | Transamerica | BungeeNeRF    | 24.608          | 24.350          | 24.357          | 24.608          | 24.415          |
> |                                  | Switch-NeRF-2 | 24.468          | 24.233          | 24.410          | 23.495          | 24.220          |
> |                                  | Switch-NeRF-4 | **25.242** | **25.276** | **25.504** | **24.672** | **25.226** |

---

> > ### Author Response · Authors · 2022-11-19
> > **Response to Reviewer QZyS (2/3)**
> >
> > >R2.Q3: Results on Block-NeRF dataset.
> >
> > We add the results on Block-NeRF dataset in the Appendix B. The Mission Bay Dataset used by Block-NeRF consists of 12,000 images with an image resolution of around 1200x900. Although the data volume is similar to the datasets used by Mega-NeRF and Switch-NeRF, the training setting of Block-NeRF is quite different from ours. The Block-NeRF uses much larger computing and memory resources in its training setting. It trains each scene block with 32 TPU v3 cores combining 512 GB memory, which is not easy to be followed by the common computer workstations. The batch-size of Block-NeRF for each block is 16384, significantly larger than the averaging batch-size 1024 for each sub-network in Mega-NeRF and Switch-NeRF. The networks of Block-NeRF are trained with full precision while ours is with bfloat16 precision. Besides, an important part of the training data, i.e., the masks for dynamic objects in the scenes used by Block-NeRF, are also not released in the available training data. The Block-NeRF does not open-source its code, which makes us difficult to align its hyper-parameters on their training data, and also blocks us from training their model on another public benchmark for comparison.
> >
> >
> > Due to our lack of comparable computing resources and the unavailability of the same training data used by Block-NeRF, it is not feasible for us to train a Switch-NeRF model based on the same setting as Block-NeRF, to allow a direct and fair comparison. We thus train our network on the Block-NeRF dataset with a similar setting used in our paper. We use 8 NVIDIA RTX 3090 GPUs to train a Switch-NeRF model with 8 experts, and sample 1664 rays on each GPU, resulting in a batch-size of 1664 on average for each expert. This is far less than the batch-size of 16384 for each Block in Block-NeRF. The width of each layer is set as 512 as in Block-NeRF. We also use the positional encoding of Mip-NeRF as used by Block-NeRF. We sample 256 points each ray for the coarse and ﬁne networks in training, and 512 points each ray for testing. We use the validation dataset of Block-NeRF to evaluate the performance with the PSNR, SSIM and LPIPS metrics. The experimental results are shown in the table below and also in Table 8 in Appendix B.
> >
> >
> > Table: Accuracy of Switch-NeRF on the San Francisco Mission Bay Dataset proposed by Block-NeRF. We train the Switch-NeRF with limited computing resources, half float precision, and much smaller batch-size compared to Block-NeRF. Besides, Switch-NeRF does not utilize the important dynamic object masks that are used by Block-NeRF, as they are not available in the released training dataset.
> >
> > | **Method**              | **PSNR$\uparrow$** | **SSIM$\uparrow$** | **LPIPS$\downarrow$** |
> > |-------------------------|--------------------|--------------------|-----------------------|
> > | Switch-NeRF (8 experts) | 23.86              | 0.762              | 0.489                 |
> >
> >
> > >R2.Q4: Video.
> >
> > Thank the reviewer for the suggestion of a qualitative video demo. We will make video demos and release them together with the codes and trained models once the paper is accepted.
> >
> > >R2.Q5: How do different experts in Switch-NeRF learn different semantic parts? What will different experts learn if Switch-NeRF deals with building-level cases?
> >
> > Thank the reviewer for the questions. Since the gating network is jointly learned with NeRF-experts in Switch-NeRF, the radiance fields encoded by an expert implicitly have internal consistency. The radiance fields, aka density and color, are physical properties of a scene, which are directly related to the semantics of the scene. Our Switch-NeRF learns the gating network essentially based on the spacial (i.e., coordinates) and appearance (i.e., density and color) distribution of 3D scene points. The former one (i.e., the 3D point coordinates) is the input of the gating network, leading to spatially closer points to be more possibly assigned into the same expert. The later one (i.e., the density and color) affects the gating network through the pixel rendering loss, leading to the points with more similar appearance to be more possibly assigned to the same expert. Besides, the distributions of density and color of 3D points are continuous in 3D scene space, which also facilitates the learning of the scene decomposition by the gating network, i.e., points with higher spatial and appearance similarity being assigned to the same NeRF expert. Therefore, the experts in Switch-NeRF can roughly specialize to different semantic parts after training.
> >
> > We additionally visualize the 3D radiance fields of a Switch-NeRF trained with 16 experts on the UrbanScene3D-Sci-Art dataset in Fig. 7 in Appendix D, and the rendering accuracy is also shown in Table 10 in Appendix D. We can observe that the NeRF experts in our Switch-NeRF network can still roughly specialize to different fine-grained semantic parts for both the scene-level and building-level architectures.

---

> > > ### Author Response · Authors · 2022-11-19
> > > **Response to Reviewer QZyS (3/3)**
> > >
> > > >R2.Q6: The improvement on different scenes.
> > >
> > > The number of experts is a factor affecting the scene modeling performance for scenes with different scales. The campus is a very large-scale scene (5871 images with 5472x3648 resolution). We add more experimental results in the table below and in Table 10 in Appendix D, to show the performance of the model with 16 experts. In our original submission, we also provided an ablation study on Building scenes in Table 3 of the paper. The results show that, for a dataset with a larger scale, more expert networks can consistently produce better rendering results.
> > >
> > >
> > > Table: The accuracy of different expert numbers. Increasing expert number can consistently improve the accuracy.
> > >
> > > | Method         | Building |       |       | Sci-Art |       |       | Campus |       |       |
> > > |----------------|----------|-------|-------|---------|-------|-------|--------|-------|-------|
> > > |                | PSNR$\uparrow$     | SSIM$\uparrow$  | LPIPS$\downarrow$ | PSNR$\uparrow$    | SSIM$\uparrow$  | LPIPS$\downarrow$ | PSNR$\uparrow$   | SSIM$\uparrow$  | LPIPS$\downarrow$ |
> > > | Switch-NeRF-8  | 21.54    | 0.579 | 0.474 | 26.52   | 0.795 | 0.360 | 23.62  | 0.541 | 0.609 |
> > > | Switch-NeRF-16 | **22.49**    | **0.625** | **0.429** | **27.17**   | **0.819** | **0.327** | **24.40**   | **0.578** | **0.553** |
> > >
> > >
> > >
> > > >R2.Q7: "Robust".
> > >
> > > Thank the reviewer for pointing this out. We agree that the "robust" is not concrete here. What we wanted to express is that a deeper gating network can guarantee enough parameters to boost the accuracy. We have changed this line in the revised paper to be "we design a deeper gating network to guarantee enough parameters to boost the accuracy of the scene rendering''. There is also an ablation study in Table 2 in our initial paper, which shows that with a deeper gating network, we can achieve better scene rendering results.
> > >
> > > >R2.Q8: Effect of two gating operations.
> > >
> > > Thank the reviewer for the suggestion. We add an additional ablation study on two gating operations in Appendix C. We train a Switch-NeRF with 2 gating networks, each acting as a gating operation. The first gating network is added after the $1$-st linear layer of the main network, and the second gating network is added after the $4$-th linear layer of the main network. The total number of expert layers is the same as that of the Switch-NeRF using 1 gating network to make them have similar network capacity. The results in terms of both accuracy and efficiency are shown in the table below and also in Table 9 in Appendix C. From the results, we can clearly see that two gating operations cost more training time, training memory, and testing time, while achieving a similar accuracy to the model with one gating operation. This suggests that the number of gating operations is not critical for the model performance, and one gating operation already shows sufficient capabilities in the dispatch of points.
> > >
> > >
> > > Table: Accuracy and efficiency with different numbers of gating operations. The Switch-NeRF with two gating operations and the same number of expert layers, does not improve the rendering accuracy over the model with one gating operation, while consuming more memory and time for training and testing.
> > >
> > > |  |   | |  | Train  |       | Test  |      |
> > > |---------------|-------|-------|-------|--------|-------|-------|------|
> > > | **Gating number** | **PSNR$\uparrow$**  | **SSIM$\uparrow$**  | **LPIPS$\downarrow$** | **Mem.**    | **Time**  | **Mem.**   | **Time** |
> > > | 1             | 21.54 | 0.579 | 0.474 | 10182M | 42.5h | 5847M | 110s |
> > > | 2             | 21.55 | 0.574 | 0.477 | 15315M | 56.2h | 5838M | 152s |

---

### Official Review · Reviewer_qZQg · 2022-10-26

**Confidence:** 4
**Correctness:** 4
**Technical Novelty And Significance:** 4
**Empirical Novelty And Significance:** 4
**Recommendation:** 8

**Clarity, Quality, Novelty And Reproducibility:**

> "the different sub-networks are typically optimized independently, and thus the inconsistency among them cannot be effectively handled during the optimization"

I am not sure about this. I think BlockNerf handles the inconsistency between NeRFS quite effectively, through the compositing/blending technique. That was one of the contributions of the work.


The three "limitations" of BlockNerf and MegaNerf described in the introduction are all essentially the same limitation -- the partitioning is handcrafted rather than learned.

> "Another problem is that common MoE implementations (Hwang et al., 2022) define a capacity factor to limit the number of tokens dispatched to each expert"

Why do you believe this is a problem? I think this is an intentional choice of that work, to help the experts specialize.



> "It is because in NeRF ..." "It is reasonable ..."

I think it's bad to start a sentence like this. (The word "It" is referring to nothing here.)


> "Multiple gating operations will remarkably influence training and testing speed."

"Remarkably" is usually good, so this sentence sounds like you are saying multiple gating operations will improve things, but I think you mean the opposite.


The section before 3.1 describe the use of a top-1 operation, but then 3.1 describes the use of top-k and averaging between their outputs. Later it says k is either 2 or 1, and later still it says k=1. It would be great to eliminate all of the ambiguity here.


> "We can put the gating network to the deeper layers"

What does this mean?

> "However, the layer numbers of expert networks will shrink and the sparsity of the whole network will be limited"

What does this mean? I don't think it's possible for the number of layers to shrink unless the implementor changes some code.

> "we put the gating network at the beginning of Switch-NeRF to maximize the layer numbers of the expert networks"

Again, this sounds very unusual. Is the gating network adjusting the number of layers in the expert NeRFs? Perhaps this is similar to differentiable neural architecture search, but I did not see any related work cited on this subject.



> "The random decomposition randomly dispatches 3D points into sub-networks. "

I am not sure that I understand this, because the method actually produces pretty good results according to Table 2. It seems like two points that are infinitesimally close to each other will be dispatched to different networks at random, and no specialization can occur. Why does this produce results so close to distance-based decomposition, and so close to Mega-NeRF (i.e., within 1 PSNR point)?

**Strength And Weaknesses:**

Using MoE here makes sense. The idea appears to be well implemented. The experiments are very thorough, and answer my main curiosities about the model details.

There are a few issues with the writing/clarity, which I list next, but overall the paper is good.


**Summary Of The Paper:**

This paper combines a mixture of experts approach with NeRFs for large-scale neural scene rendering. The key idea is to learn a gating network which uses the positional embedding to choose which NeRF model will be queried for the density and color of a given point. All NeRFs share the same head. The gating network needs to be regularized, similar to other MoE models.


**Summary Of The Review:**

I think this is a good paper, using MoE as a natural learnable alternative to handcrafted assignment strategies like those in BlockNerf and MegaNerf. The experiments answer a variety of interesting questions. The writing could be improved, but overall things make sense.

---

> ### Author Response · Authors · 2022-11-19
> **Response to Reviewer qZQg**
>
> We thank the reviewer for the positive feedback and insightful comments. We address the detailed questions and comments below.
>
> >R1.Q1: Handling inconsistency.
>
> The Block-NeRF and Mega-NeRF also handle the inconsistency. The Block-NeRF directly splits the entire dataset into different blocks, based on the spatial distribution of the images in the dataset. It defines a 50% overlap for each pair of adjacent blocks to maintain consistency between adjacent blocks in training. During inference, the composition of the results from different NeRF-Block is performed by post-interpolation on the 2D image plane to achieve a transition consistency crossing blocks. The Mega-NeRF partitions training images by viewing rays, aka pixels. A training pixel can be assigned to several parts. During inference, it also sets a predefined 15% overlap for neighboring parts. Therefore, it also handles the inconsistency to some extent. Both these two methods require hand-crafted rules (e.g., overlap ratios, distance thresholds for different blocks/parts, and interpolation strategies) in the training and inference to achieve a better consistency. In contrast to these works, Switch-NeRF learns an end-to-end large-scale scene rendering network, through jointly performing the scene decomposition, the NeRF-expert modeling for each decomposition, and the scene fusion with a unified prediction head. The proposed network structure enables us to handle the inconsistency problem implicitly via directly optimizing the whole network, without applying human-defined scene-specific partitioning and fusion rules.
>
> >R1.Q2: Three limitations of Block-NeRF and Mega-NeRF.
>
> Yes, each of these three limitations of Block-NeRF and Mega-NeRF is related to one another, which is essentially caused by the handcrafted scene partitioning.
>
> >R1.Q3: Capacity factor in MoE.
>
> The problem with the capacity factor is from observations during our implementation and experiments on the targeted task, i.e., NeRF for large-scale scene modeling. The commonly-used general implementations of MoE usually use the capacity factor during both the training and testing. However, we observed that using capacity factor during testing largely decreases the rendering quality. Based on this observation, we implement the testing codes by disabling the capacity factor, instead of following the common practice.
>
> >R1.Q4: The word "It" and "remarkably".
>
> Thank the reviewer for the advice. We have fixed the corresponding sentences in the revised paper to make them more clear.
>
> >R1.Q5: Gating description.
>
> Thank the reviewer for the advice. We have fixed k to 1 in the description in Sec 3.1, and have also rewritten the equation to eliminate ambiguity.
>
> >R1.Q6: Gating network and the number of layers in the expert.
>
> Thank the reviewer for the comments. The motivation of our paper is to increase the capacity of the NeRF network while maintaining almost constant computational cost for each sample which defines the overall network depth. "We can put the gating network to the deeper layers" means that placing the gating network closer to the prediction head of the Switch-NeRF network. This makes each sample point share more layers before the gating network. In this case, the capacity of the whole network shrinks, as the network capacity is controlled by the number of layers of each unshared expert, when a constant computational cost for each sample is defined. Therefore, we chose to place the gating network at the beginning of the whole network. We have modified the text in the revised version to make this point more clear.
>
> >R1.Q7: Explanation of Random decomposition and its results.
>
> The random decomposition uniformly dispatches 3D points into different sub-networks in both training and testing. This uniform sampling makes each sub-network actually learn a representation of the whole scene after the training. The final network with larger capacity can be seen as an ensemble of several smaller sub-networks. Therefore, the random decomposition can still obtain satisfactory results, but it performs worse compared to the original NeRF, the distance-based decomposition, and Mega-NeRF, especially in terms of the structural similarity metric (i.e., SSIM) and the perceptual similarity metric (i.e., LPIPS). This also suggests that random decomposition without specialization cannot model detailed scene structures well.

---

> > ### Comment · Reviewer_qZQg · 2022-11-22
> > **OK**
> >
> > It sounds like these responses agree with me, but my hope was that the PDF would get updated, not merely that the problems would be acknowledged. It seems like most of these issues are still present ("inconsistency" claim about blocknerf, three limitations being the same, "capacity factor" described as a "problem" instead of a solution, "It is reasonable" and other sentences starting with an ambiguous "It is...", top-1 vs top-k mixups). Overall I still like the paper but this is a bit disappointing.

---

> > > ### Author Response · Authors · 2022-11-23
> > > **R1.Q8: Revision of the paper.**
> > >
> > > Thank the reviewer for the helpful comments on our paper. Regarding the questions of "inconsistency of BlockNeRF", "three limitations", and "capacity factors", we are sorry that we considered them as questions for discussions, and did not realize that we should accordingly update the paper to add the discussions for these questions. We hope that our elaborations about them in the rebuttal are satisfactory to the reviewer, and we will carefully merge those discussions into the final revision. For the two "it is" sentences specifically mentioned by the reviewer, we addressed them in the previous paper revision, respectively in the first paragraph of Sec. 3.1 and in the paragraph of "Model Analysis: Efficiency" in Sec. 4.4. We will carefully check the whole paper, to fix any similar presentation issue related to "it is" in our final revision. For the mixup of "top-1" and "top-k", we changed both our method formulation and description to "top-1" in Sec. 3 in the previous revision, following the suggestion of the reviewer. There are two other spots in the paper where we discussed the top-k choice, which may be possibly related to what referred to the mixup by the reviewer. The first spot is in Sec. 2 (i.e. Related Works), where we introduce the "top-k" choice of other classic MoE methods, instead of our design choice. The other spot is in our ablation study in Sec. 4.4, i.e., "Model Analysis: Top-k and capacity factor". In this part, we performed an ablation study to show the performance differences between top-1 and top-2. To be more specific, we will directly change sub-title of this part to "Effect of top-2 and capacity factor" in the final revision, and also modify the caption of Table 4 accordingly, to eliminate any possible ambiguity from wordings. Thank the reviewer again for the helpful comments for further improving the paper.

---

### Author Response · Authors · 2022-11-19
**Paper changes**

- Add the training iteration number in Sec 4.2.
- Add several references.
- Fix writing issues raised by reviewers.
- Use Top-1 in the MoE formulation in Sec. 3 to eliminate the ambiguity.
- Add experiments on BungeeNeRF datasets in Appendix Sec. A and Table 7.
- Add experiments on Block-NeRF datasets in Appendix Sec. B and Table 8.
- Add experiments on two gating operations in Appendix Sec. C and Table 9.
- Add more experiments on expert numbers in Appendix Sec. D, Table 10 and Figure 7.
- Add experiments of fast rendering in Appendix Sec. E, Table 11 and Figure 8.

---

### Comment · Area_Chair_Dwdc · 2023-03-22
**Open-source Code**

Dear Authors:

Congratulations again on your paper getting accepted by ICLR!

AC has recently received multiple requests on this paper for reproducing the results.

Could authors give a concrete timeline for open-source code?

Thanks

AC

---

> ### Author Response · Authors · 2023-03-22
> **Beta release of the code**
>
> Dear AC and the community,
>
> Thank you for your reminder about the open-source of our code. We appreciate the interest from the community in our work.
> In response to the requests of the community, we decide to release a beta version of our code in this [link](https://github.com/MiZhenxing/Switch-NeRF). Please note that there are still some bugs and the documentation is currently not complete. We will continue improving the code and adding necessary documentation and trained models.
>
> In terms of a concrete timeline, we expect to release a more stable version of the code with complete documentation in one week.
>
> Thank you for your patience and understanding.
>
> Best regards,
>
> Zhenxing Mi

---

### Decision · Program_Chairs · 2023-01-20

**Decision:**

Accept: poster

**Justification For Why Not Higher Score:**

1. Not very efficiency
2. Hard to adapt to new environment

**Justification For Why Not Lower Score:**

Good and useful engineering efforts.

**Metareview: Summary, Strengths And Weaknesses:**

Four experts reviewed this paper with all accepted recommendations. The area chairs agree that this work makes a very important contribution by combining a mixture of experts' approaches with NeRFs for large-scale neural scene rendering. The reviewers did raise some valuable concerns that should be addressed in the final camera-ready version of the paper. The authors are encouraged to make the necessary changes and include the missing references in the final version.

**Note From Pc:**

if the above contains the word "oral" or "spotlight" please see: "oral" presentation means -> notable-top-5% and "spotlight" means -> notable-top-25%. As stated in our emails, we are disassociating presentation type from AC recommendations